# Phenotypic drug susceptibility testing for *Mycobacterium tuberculosis* variant *bovis* BCG in 12 hours

Buu Minh Tran ⓘ, Jimmy Larsson, Anastasia Grip ⓘ, Praneeth Karempudi ⓘ & Johan Elf ⓘ ✉

Drug-resistant tuberculosis (DR-TB) kills ~200,000 people every year. A contributing factor is the slow turnaround time (TAT) associated with drug susceptibility diagnostics. The prevailing gold standard for phenotypic drug susceptibility testing (pDST) takes at least two weeks. Here we show that growth-based pDST for slow-growing mycobacteria can be conducted in 12 h. We use *Mycobacterium tuberculosis* variant *bovis* Bacillus Calmette-Guérin (BCG) and *Mycobacterium smegmatis* as the mycobacterial pathogen models and expose them to antibiotics used in (multidrug-resistant) tuberculosis (TB) treatment regimens - *i.e.*, rifampicin (RIF), isoniazid (INH), ethambutol (EMB), linezolid (LZD), streptomycin (STR), bedaquiline (BDQ), and levofloxacin (LFX). The bacterial growth in a microfluidic chip is tracked by time-lapse phase-contrast microscopy. A deep neural network-based segmentation algorithm is used to quantify the growth rate and to determine how the strains responded to drug treatments. Most importantly, a panel of susceptible and resistant *M. bovis* BCG are tested at critical concentrations for INH, RIF, STR, and LFX. The susceptible strains could be identified in less than 12 h. These findings are comparable to what we expect for pathogenic *M. tuberculosis* as they share 99.96% genetic identity.

Tuberculosis (TB) has afflicted humans for millennia and remains a major public health threat[1]. As COVID-19 fatalities decrease, *Mycobacterium tuberculosis* (*Mtb*), the causative agent of TB, has reclaimed its position as the primary infectious cause of death worldwide. An estimated 10.8 million people contracted TB in 2023, and in the same year, a total of 1.25 million died from the disease[2]. A major driver of TB mortality is the increasing prevalence of drug-resistant TB (DR-TB), multidrug-resistant TB (MDR-TB), pre-extensively drug-resistant TB (pre-XDR-TB), and extensively drug-resistant TB (XDR-TB) infections[3–8]. In 2022, nearly half a million people developed multidrug-resistant or rifampicin-resistant TB (MDR/RR-TB)[2,9]. In 2019, around 25% of global deaths from antimicrobial-resistant infections were attributed to rifampicin-resistant TB[9]. Additionally, an estimated 1 million cases of TB resistant to isoniazid (another essential first-line drug, without simultaneous rifampicin resistance) emerged in the same year[9].

Treating drug-resistant TB, particularly XDR-TB, requires lengthy, expensive drug regimens and is often linked to worse treatment outcomes, including a higher mortality rate, compared to susceptible cases[10–13]. In a multicentre cohort study, more than half of the misdiagnosed drug-resistant TB cases received inadequate therapy resulting in death[14]. Therefore, an efficient TB treatment regimen requires a combination of antibiotics to minimize treatment failure due to resistance and the emergence of multidrug-resistant TB. More importantly, it is recommended that drug susceptibility tests (DST) are used to guide initial treatment selection[15,16].

Presently, both genotypic and phenotypic drug susceptibility testing (gDST and pDST) methods are employed in combination to accurately determine the drug susceptibility of *Mtb*. Genotypic methods like nucleic acid amplification tests (NAAT) and line probe assays (LPA) are widely used. The Xpert® MTB/XDR NAAT (Cepheid), which

Department of Cell and Molecular Biology, SciLifeLab, Uppsala University, Uppsala, Sweden. ✉e-mail: johan.elf@icm.uu.se

detects resistance to isoniazid (INH), fluoroquinolones (FQ), ethionamide (ETH), and second-line injectable drugs (SLIDs), can detect 16 resistance mutations in TB from sputum in under 90 minutes[17]. The LPA Genotype MTBDRplus (Hain Lifescience GmBH) can detect RIF- and INH-associated mutations with high specificity in less than 6 h[18]. Despite the reported short TAT of gDST in an optimal research laboratory, the time to result under operational conditions ranges from one to ten working days[8]. Whole genome sequencing (WGS) is also recommended in gDST, as it can detect resistance mutations outside the limited spectra of NAAT and LPA[18–20]. Although WGS gDST methods are generally effective for *Mtb*, they are expensive, inaccessible to low-resource areas, limited in sensitivity for heteroresistant infections, incapable of detecting resistance to the new drugs, *e.g.*, bedaquiline, delamanid, *etc*[21,22]. Moreover, although WGS or genotypic predictions of susceptibility of *Mtb* were observed to be correlated with phenotypic susceptibility to first-line drugs[23–25], it is unlikely that WGS will soon completely replace phenotypic DST for TB in many countries[26].

Phenotypic drug susceptibility testing (pDST) using liquid or solid media remains the gold standard DST method. Liquid culture-based pDST such as BACTEC MGIT 960 (Becton Dickinson) produces TB diagnostic results within two weeks, likely the fastest commercially available option in clinical use[8,18]. pDST on solid media is simple and accessible, but the TAT is more than 21 days[21]. The long TAT for pDST was observed to result in acquired drug resistance, thereby affecting the efficacy of TB treatment[27]. Despite the limitations arising from the slow growth of *Mtb*, pDST remains dependable and proficient in detecting drug resistance resulting from unidentified mechanisms overlooked by genotypic analysis[8]. A nanomotion technology detecting the vibration of bacterial cells attached to cantilevers has been used for drug susceptibility testing of *Mtb*, providing results in 21 h[28], starting from a culture-positive sample. However, this technology is expensive, and the instrument can only be used to test one sample with one antibiotic at a time.

Our combination of microfluidics and microscopy has recently been demonstrated to speed up growth-dependent pDST[29,30] for UTI pathogens and was awarded the Longitude Prize for antimicrobial resistance (AMR) in 2024[31]. With this methodology, bacteria are trapped in thousands of microchannels on a microfluidic chip. Different areas of the chip are exposed to different media, making it possible to study the effect of antibiotic drugs compared to untreated control. The setup allows for highly controlled experiments where growth impact is measured from the length extension of individual cells instead of waiting for the cells to multiply. Our previous study focused on uropathogenic *Escherichia coli*. We could produce pDST results of clinical samples in less than 30 minutes, with sensitivity and specificity of 86−100% compared to a conventional disc-diffusion[29]. For these tests, low bacterial counts ($10^4$ CFU/mL) in a spiked sample were sufficient. In the following study, we tuned the method to detect mixed-species samples (*e.g., E. coli, Klebsiella pneumoniae, Staphylococcus aureus*, and *Pseudomonas aeruginosa*) by combining the pDST with fluorescence in situ hybridization (FISH) to identify each species following susceptibility testing. A deep learning algorithm was embedded in the analysis pipeline to increase the cell segmentation robustness[30].

A few groups have used microfluidics to study the physiology of mycobacterial cells. Aldridge et al., loaded *M. smegmatis* into microfluidic chips to study their asymmetric growth at the single-cell level and test the differential susceptibility to antibiotic stress[32]. Wakamoto et al., also used microfluidic culture and time-lapse microscopy to study the dynamic persistence of *M. smegmatis* under antibiotic stress with isoniazid[33]. Baron et al., used a microfluidic acoustic-Raman platform to assess the impact of isoniazid on *M. smegmatis* in real time[34]. Another microfluidic device was developed by Wang et al., to visualize the long-term growth of *M. smegmatis* under antibiotic

pressures[35]. In yet another study, Mistretta et al. attempted to create microfluidic platforms to track the drug response dynamics in single mycobacterial cells[36,37]. *M. tuberculosis* was loaded into a microfluidic chip by Chung et al., to investigate the biology of growth at the single-cell level[38].

Another reference point is rapid pDST based on direct imaging of various bacterial strains in agarose channels, including *Mtb*[39–42]. The system, called QuantaMatrix microfluidic agarose channel (QMAC), was integrated with MGIT liquid culture, resulting in a short TAT of the culture identification (-2 weeks) and subsequent pDST from *Mtb* clumps (-1 week)[42].

In this study, we present a rapid pDST approach for *M. bovis* BCG and *M. smegmatis* as models for slow-growing tuberculous and fast-growing nontuberculous pathogens. The approach combines microfluidics, single-cell imaging, and deep neural network (DNN)-based image analysis.

## Results

### Fast phenotypic drug susceptibility testing (pDST) on a microfluidic chip

The workflow for the fast pDST for *M. bovis* BCG is shown in Fig. 1a. We start with a liquid culture sample. The cells in the liquid culture are loaded into the microfluidic device for compartmentalization in microchambers. The growth of the cells in the microchambers under different antibiotic exposures is monitored by phase-contrast microscopy. The resulting image data is segmented with a deep neural network (DNN)-based method.

Figure 1b shows the microfluidic chip design. Details of the chip design and fabrication are given in the Supplementary Information (SI) and Supplementary Fig. 1. The chip has two rows of microchambers that function as cell traps; there are 100 microchambers with dimensions of $50 \times 60 \times 1\,\mu m$ on each row. At the end of each microchamber, constrictions down to 300 nm prevent the cells from escaping into the back channel while allowing a constant media flow over the cells. Trapping mycobacteria in microchannels of our typical mother machine-like chip[29,30,43] was unreliable due to their tendency to form clumps. Once the cells have been loaded into the microchambers, their growth is monitored by phase-contrast time-lapse imaging. To find the outlines of single cells in each image frame, we use the morphology-independent segmentation algorithm of Omnipose[44] (Fig. 1c). We describe the growth rate estimation and normalization details in Fig. 1d–i.

In this work, *M. bovis* BCG and *M. smegmatis* are used as models for tuberculous and nontuberculous pathogens. Due to the difference in their growth rate, we applied the measurement time of 3 + 24 h and 1 + 3 h for *M. bovis* BCG (Fig. 1d–f) and *M. smegmatis* (Fig. 1g–i) respectively; the former terms are the times when media without drugs are supplied to both rows of microchambers, and the latter terms are the times when the drug is supplied to the medium in the treatment row. The expansion of the total cell area was monitored for 27 h (*M. bovis BCG*) and 4 h (*M. smegmatis*). The expansion is due to the cells' growth within the microchambers. Figure 1d shows the total areas of *M. bovis* BCG plotted as a function of time for the untreated reference population (left) and treatment population (right; isoniazid (INH) 0.5 mg/L). Each line in Fig. 1d presents data from an individual microchamber. Images were captured every 10 min. The drug was added to the treatment population after 3 h (dashed red line). The growth rates were estimated with a sliding window of 3 h for *M. bovis* BCG, and thus, the growth rates are plotted from 3 h after starting the experiment (Fig. 1e). Figure 1f shows the growth rate of *M. bovis* BCG in INH 0.5 mg/L normalized to the reference population. This type of normalization has previously been used[29,30] to reduce the isolate dependence in the antibiotic response. The area growth and growth rates of the reference (left) and treatment population (right: rifampicin (RIF) 10 mg/L) of *M. smegmatis* are shown in Fig. 1g, h. The pDST profile of *M. smegmatis* in

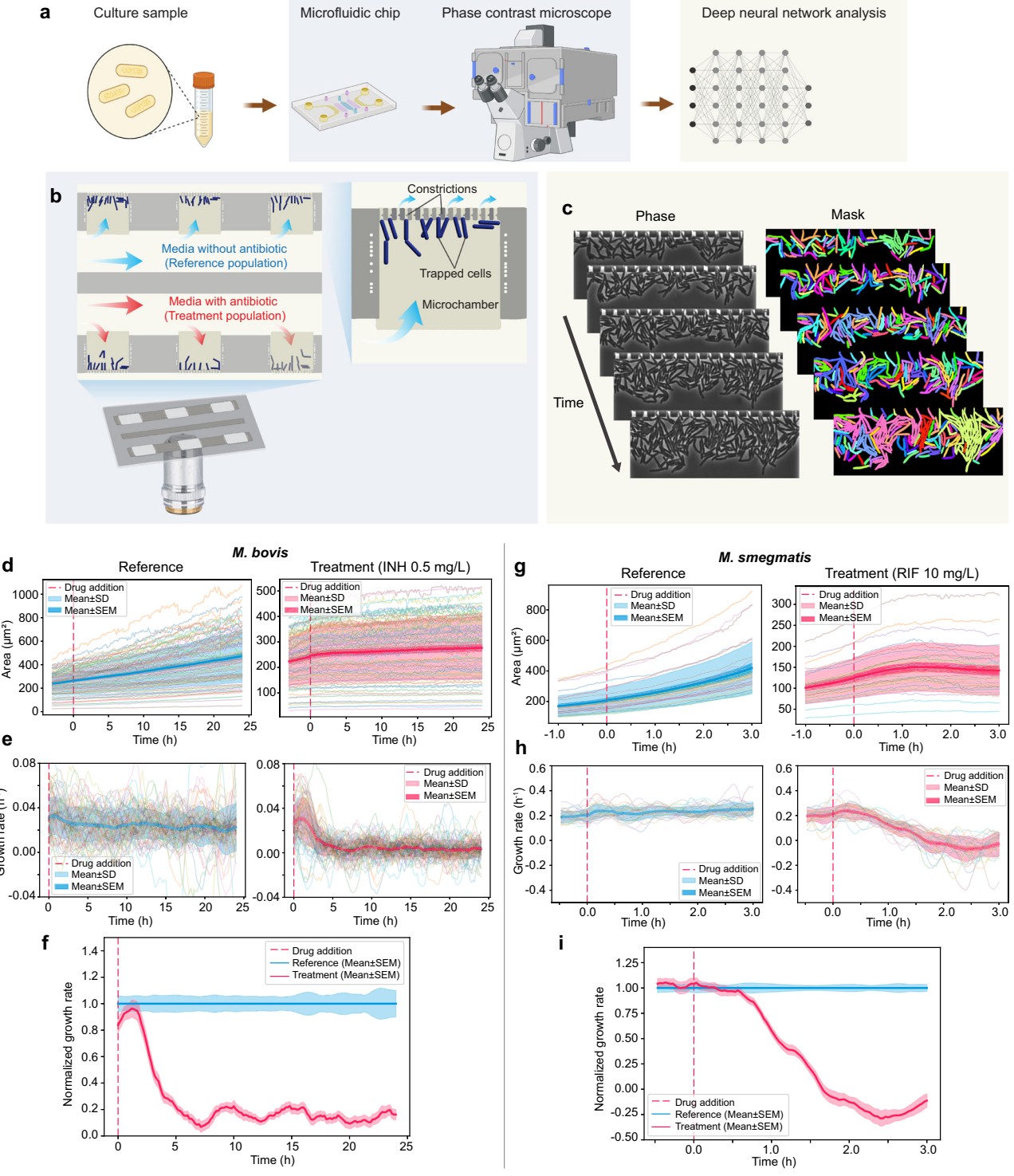

**Fig. 1 | Schematic representation of the fast pDST workflow for *M. bovis* BCG and *M. smegmatis*. a** Main steps in the workflow (Created in BioRender. Lab, E. (2025) https://BioRender.com/e13c692). **b** Illustration of the microchamber cell traps. **c** Analysis of time-lapse stacks using Omnipose for cell segmentation. **d–i** Step-by-step growth rate calculation and normalization for *M. bovis* BCG (**d–f**) and *M. smegmatis* (**g–i**). **d**. Total areas of *M. bovis* BCG [Reference (left) and Treatment population (right; isoniazid INH 0.5 mg/L)]. Each line is data from an individual microchamber. The dashed line indicates the drug addition time. **e** *M. bovis* BCG growth rates. **f** *M. bovis* BCG growth rate normalization (pDST profile) at INH 0.5 mg/L (mean ± SEM). **g–i** Corresponding area growth (**g**), growth rates of *M. smegmatis* (rifampicin RIF 10 mg/L) (**h**), and its pDST profile at RIF 10 mg/L (**i**).

RIF 10 mg/L is shown by normalized growth rates in Fig. 1i. In the case of *M. smegmatis*, we measured at 2 min intervals and growth rates were calculated with a sliding window time of 30 min. The occasional negative growth rate indicates shrinkage of the cell area, which can, for example, be due to lysis.

## Data analysis using a deep neural network

Accurate cell segmentation tools for mycobacteria are not available. Single-cell tracking studies on mycobacteria have so far been performed using manual annotation, likely due to their small and varied cell size and the abnormally asymmetric aging and division[32,45]. An

artificial neural network architecture, U-net[46], has recently been used to segment TB cords from lens-free microscopy images[47]. We used a related segmentation algorithm called Omnipose[44] to segment cells. Omnipose has previously achieved high accuracy and morphological independence in the segmentation of mixed bacterial samples, antibiotic-treated, elongated or branched cells[30,44], and parallel to our work, it was used for segmenting snapshots of *M. tuberculosis*[37].

We created a 120-image ground truth training dataset of mycobacterial cells under diverse experimental conditions, encompassing *M. bovis* BCG and *M. smegmatis* captured by two different cameras (Methods) on agarose pads and in microchambers, including cells subjected to antibiotic treatment. High-density mycobacterial clumps or cords in microchambers were membrane stained using 3HC-3-Tre dye[48] to assist the annotation (Fig. 2a). Mycobact_1 to Mycobact_4 were trained from scratch, while Mycobact_5 and Mycobact_6 were fine-tuned based on the Omnipose default model for bacteria in phase contrast. The training parameters are shown in Supplementary Table 1. The segmentation performance of the new models was evaluated by matching the segmented masks to the ground truth masks at various matching precision thresholds. Quantitatively, we assessed the model performance by comparing their average Jaccard Index[49] as a function of the intersection over union (IoU) threshold on the same dataset (Fig. 2b, c). IoU scores range from zero to one, with scores greater than 0.8 indicating when masks are visually indistinguishable from ground truth, according to human experts[44]. In Fig. 2b, c, the performance of all new models did not remarkably vary with the IoU threshold less than 0.75; with the IoU threshold greater than 0.75, the training-from-scratch models outperformed the fine-tuned models. The difference in performance suggests that the images of mycobacteria are significantly different from the cells that Omnipose was trained on before.

It should be noted that the general performance evaluation used an image dataset from two cameras with different pixel sizes, owing to our experiments being conducted on two different microscopes. However, the performance of the newly trained model significantly improved when using images from a single camera (Fig. 2b, Mycobact_2*). This improvement can be attributed to the use of more accurate parameters for the images from one camera, such as diameter and minimal cell size.

Overall, the segmentation performance of the new models showed remarkable improvements compared to the default bacteria model from Omnipose, particularly on low cell density data (Fig. 2b and d–g). However, high cell density data (*e.g.*, clumps or cords) were problematic due to overlapping cells and difficulty distinguishing single-cell boundaries. Instead of identifying single cells, the entire blob of cells was segmented as one (Fig. 2c and h, i). Movies showing image segmentation performance are supplied in the Supplementary videos (Supplementary videos 1 and 2).

## Fast detection of response to drug treatment

We used the fast pDST to determine the growth rate of *M. bovis* BCG Pasteur and *M. smegmatis* (NCTC 8159 and mc² 155) in response to three first-line drugs, rifampicin (RIF), isoniazid (INH), and ethambutol (EMB), and a third-line drug linezolid (LZD). MIC values of *M. smegmatis* and *M. bovis* BCG Pasteur were determined using the microplate-based Alamar Blue assay (MABA)[50,51] and EUCAST broth microdilution assay (BMDA)[52] (Supplementary Table 2), respectively. Figure 3a–d show the pDST profiles of *M. bovis* BCG Pasteur tested at 10x MIC of RIF (0.6 mg/L), and 1x MICs of INH (0.5 mg/L), EMB (2.5 mg/L), and LZD (0.5 mg/L), respectively. We could measure statistically significant growth rate differences between the treated (treatment) and control (reference) populations in under 3 h. Given that *M. bovis* BCG Pasteur has a replication time of 18−24 h[53], a difference in growth rate between the treatment and reference population could be detected at 1/8 to 1/6 of a generation time for slow-growing tuberculous mycobacteria (*i.e.*, *M. bovis* BCG). Data shown in Fig. 3a–d is from one biological

replication and is reproducible in at least two biological replications (Supplementary Fig. 2). At 1x MIC of RIF (0.06 mg/L), the growth rate difference between two populations was observed after around 8 h (Supplementary Fig. 2b).

For fast-growing mycobacteria, the difference in growth rates between the two populations could be detected in under 1 h (Fig. 3e–l). The average doubling time of *M. smegmatis* is 3 h[32,54,55], which means that the growth-rate difference could be detected after 1/3 of a generation. We show the data of the *M. smegmatis* measurements in at least two biological replications in Supplementary Fig. 3. We also present repeated experiments showing non-normalized growth rates as a function of time in Supplementary Figs. 4, 5, and 6. The response curves are reproducible for the same antibiotic, but the shapes of the curves differ for different antibiotics, implying that the response time is limited by the biology of cells rather than limitations in the measurement. The cases where the growth rate of the treatment population, or both the treatment and reference populations, increases rapidly and transiently just after drug addition are likely due to a pressure drop when changing media, causing a swift increase in cell area (Fig. 3a, d, and l).

## Bedaquiline requires more chemically inert tubing for the microfluidic setups

For BDQ we observed an initial impact on growth of *M. smegmatis* NCTC 8159 and *M. bovis* BCG in BDQ treatment on microfluidic chips followed by a rapid recovery (Supplementary Fig. 7a–d). In contrast, the antibiotic was effective and inhibited the growth indefinitely in a 96-well plate and the MIC values for *M. smegmatis* NCTC 8159 and *M. bovis* BCG were determined to 0.05 mg/L and 0.125 mg/L, respectively (Supplementary Table 2). We note that the antibiotic resided in the media supply tubing for a long time before reaching the fluidic chip. However, when a freshly-made tube with medium and antibiotic was switched in after the recovery of the growth rate, another period of the growth inhibition occurred (Supplementary Fig. 7c). Similarly, the recovery was also observed on *M. bovis* BCG WT in 4X MIC of BDQ (Supplementary Fig. 7d). This observation suggested that BDQ adsorbed to the tubing. Indefinite inhibition by BDQ (2.5 mg/L, 50X MIC) to *M. smegmatis* was only observed when we changed to a more chemically inert fluorinated ethylene propylene (FEP) tubing (Supplementary Fig. 7e and Supplementary video 3). Here, BDQ was effective on susceptible *M. smegmatis* already at 1X MIC (0.05 mg/L) (Supplementary Fig. 7f).

## Detection of drug-resistant strains

Just because a susceptible strain responds quickly to the drugs does not mean that a resistant strain does not show any response in this time scale. For example, the impact of an antibiotic may differ in the fluidic chip compared to the reference conditions used for DST. Additionally, a resistant strain might initially respond like a susceptible strain but later recover its growth rate[33,56–58]. Therefore, we need to test whether the concentration that affects a susceptible strain has significantly less impact on resistant strains with relevant MIC values. Additionally, we must monitor the cells for a long enough period to capture any potentially induced resistance. For this reason, we conducted the pDST on RIF, INH, and LZD-resistant *M. smegmatis* strains (Fig. 4), which were laboratory-evolved and characterized[59]. These strains harbored genomic mutations in the genes of *rpoB* (RIF resistance, MIC ≥ 50 mg/L), *inhA* and *katG* (INH resistance, MIC > 100 mg/L), and *oxiR* (LZD resistance, MIC ≥ 12.5 mg/L)[59]. The MIC values of the tested strains are shown in Supplementary Table 2. The pDST profiles of the drug-resistant strains were compared with the susceptible parental strain *M. smegmatis* NCTC 8159 (Fig. 3e, f, and h).

Figure 4a–c show the pDST profiles of *M. smegmatis* RIF E1 to E3. The growth of the treatment population of RIF E1 and E2 strains is comparable with the untreated reference population at 50 mg/L RIF

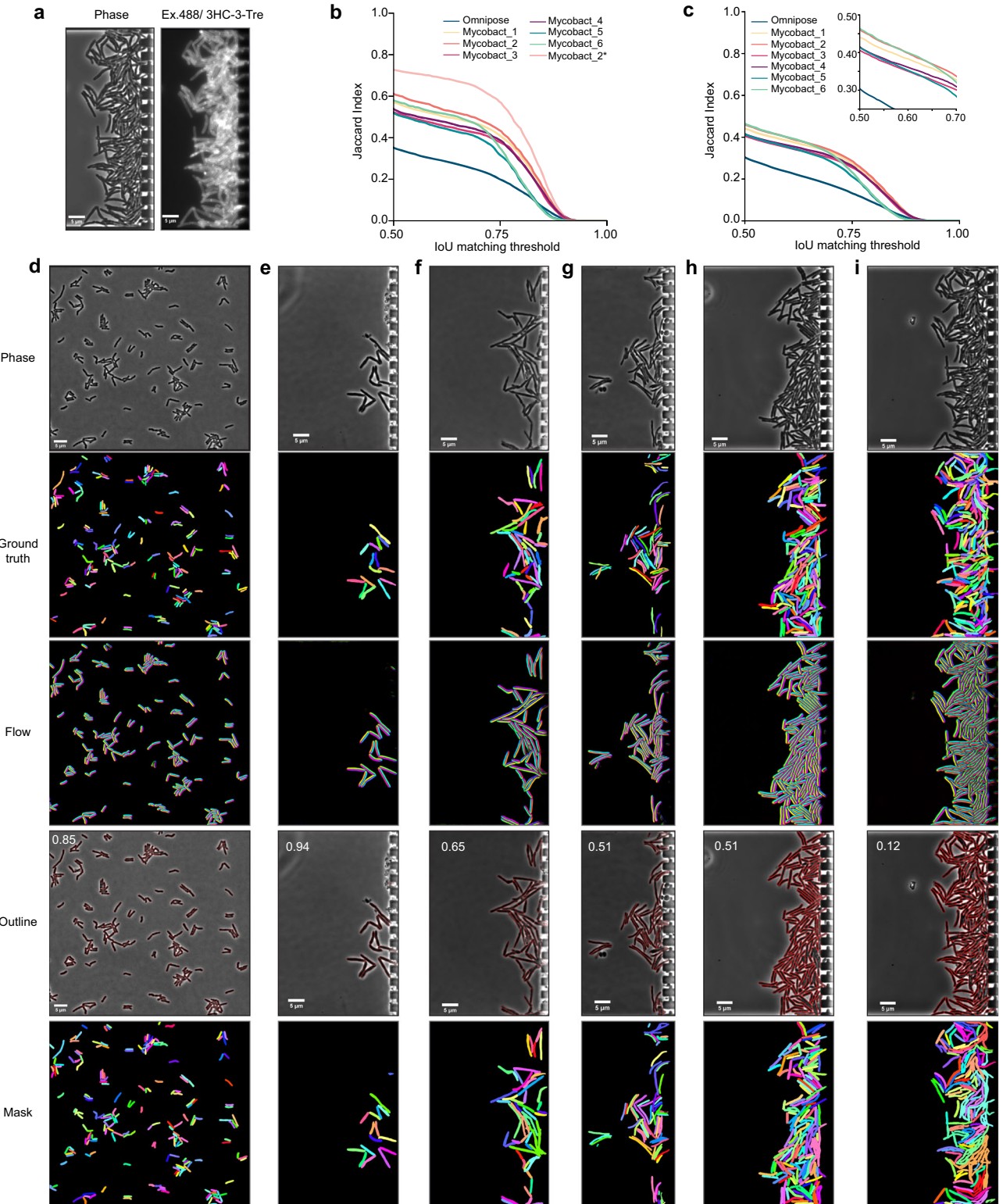

**Fig. 2 | Training new models for mycobacterial cell segmentation. a** Phase contrast image and the corresponding cell membrane image stained by 3HC-3-Tre dye. **b**, **c** Segmentation performance of different training parameters using data excluding (**b**) and including (**c**) high-density cell microchambers. Mycobact_2* in (**b**) is the Mycobact_2 model performed using images from one camera. Inset in (**c**) is a zoom-in of the *x* and *y*-axis. **d**–**i** Representative micrographs of mycobact_2

model for cells in various conditions−(**d**) *M. smegmatis* on agarose gel, (**e**) and (**f**) *M. smegmatis* in microchamber, (**g**) *M. bovis* BCG in microchamber, (**h**) and (**i**) Relatively and highly dense *M. smegmatis* clumps. Ground truth is labeled from phase images. Flow field is an intermediate output from the neural network. Average precision at an IoU threshold of 0.5 (AP@0.5) for the entire image is reported on the top left corner of the outline - scale bar 5 μm.

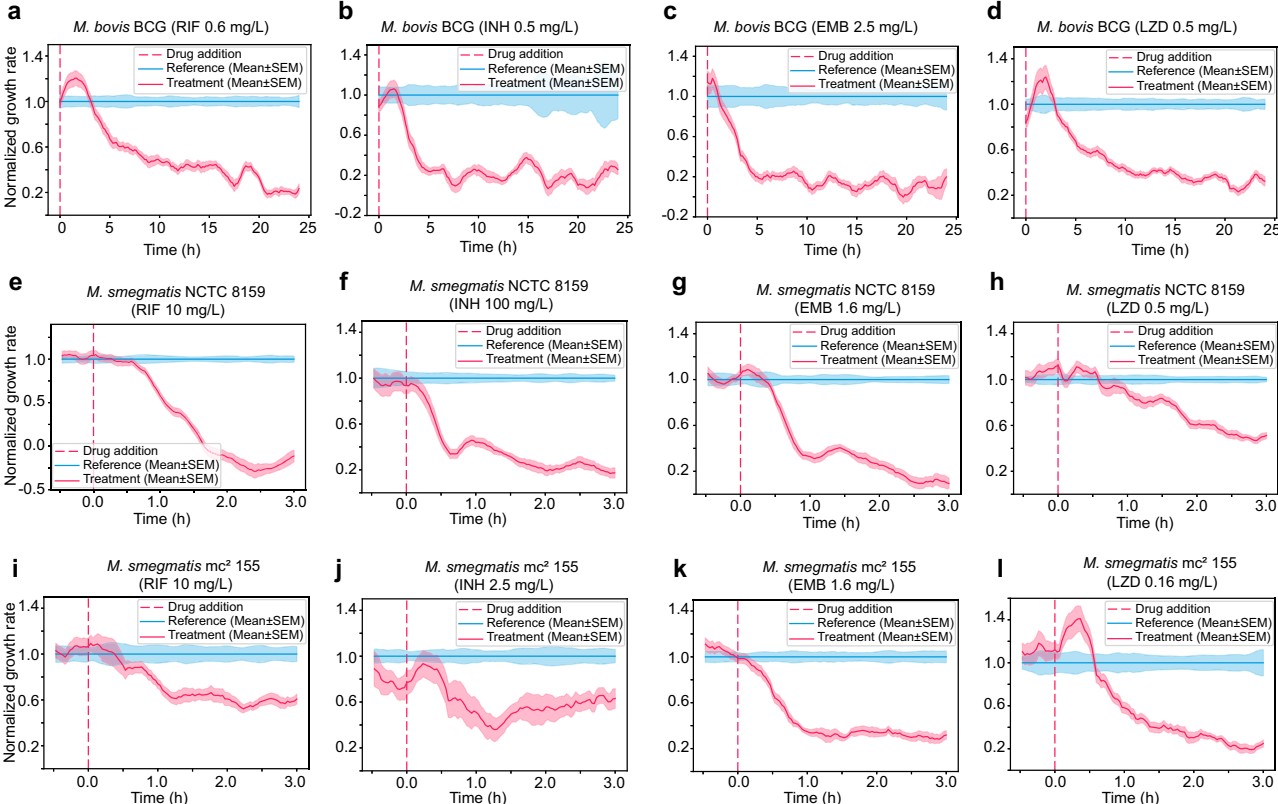

**Fig. 3 | Fast detection of response to drug treatment.** pDST assays detecting the fast response of susceptible *M. bovis* BCG Pasteur to (**a**) rifampicin (RIF), (**b**) isoniazid (INH), (**c**) ethambutol (EMB), and (**d**) linezolid (LZD); of susceptible *M. smegmatis* NCTC 8159 to (**e**) RIF, (**f**) INH, (**g**) EMB, and (**h**) LZD; and susceptible *M.* *smegmatis* mc² 155 to (**i**) RIF, (**j**) INH, (**k**) EMB, and (**l**) LZD. Data in each graph is from one biological replication. Multiple biological replication data are shown in Supplementary Fig. 2 and Supplementary Fig. 3.

concentration (Fig. 4a, b). We noted that 50 mg/L RIF is 5 times higher than the concentration that severely impacted the parental strain *M. smegmatis* NCTC 8159 (Fig. 3e). RIF E3 is less resistant and displays growth impact at 10 mg/L of RIF (Fig. 4c), but not as much as the parental susceptible strain (Fig. 3e). These results are in agreement with the 48 h REMA reference assay (Supplementary Table 2). For INH-resistant strains, the growth of the treatment population of *M. smegmatis* INH E1 is indistinguishable from the untreated reference population implying high resistance (Fig. 4d). INH E2 and INH E3 dropped in growth rate during the first 30 min after the drug was added and then gradually recovered reaching the reference level in 3 h (Fig. 4e, f). At 0.5 mg/L LZD concentration, the growth rates of the three LZD-resistant strains were not affected by the antibiotic (Fig. 4g–i) as opposed to the susceptible parental strain (Fig. 3h). We show the data of the measurements on resistant strains in multiple biological replications in Supplementary Fig. 8. Additionally, repeated experiments showing non-normalized growth rates as a function of time in response to drugs are presented in Supplementary Fig. 9. Overall, resistant and susceptible *M. smegmatis* strains could reliably be separated within 3 h by the fast pDST assay.

We also conducted tests on a panel of antibiotic-susceptible and -resistant slow-growing mycobacterial strains with attenuated virulence (*M. bovis* BCG Russia)[60]. The panel was created to provide safe quality control reagents for detecting drug-resistant *Mtb*[60]. We tested the panel using critical concentrations (CCs) for DST of drugs used in treating (drug-resistant) TB recommended by WHO[61–63] (rifampicin (RIF), streptomycin (STR), and levofloxacin (LFX)) and CLSI[64] (isoniazid(INH)).

The CC of INH in MGIT media set by CLSI is 0.4 mg/L, but we chose 0.5 mg/L to match the INH MIC value of *M. bovis* BCG Pasteur. At 0.5 mg/L of INH, the growth of *M. bovis* BCG Russia WT was

dramatically inhibited (Fig. 5a), whereas the corresponding INH-resistant strain was only slightly affected and regained growth to the level of the reference population after 12 h (Fig. 5b).

The CCs in MGIT liquid culture for RIF, STR, and LFX set by WHO are 1 mg/L. The differences in growth rate between the treatment and reference population of *M. bovis* BCG Russia WT in 1 mg/L of RIF, STR, and LFX are observed around 6 h after the drug addition (Fig. 5c, e, and g). The RIF- and STR-resistant strains were not remarkably affected by the drugs at this concentration (Fig. 5d and f), and the fluoroquinolone (FQ) resistant strain was marginally inhibited by LFX 1 mg/L in around 6 h and recovered afterward (Fig. 5h). Data from two biological replications of susceptible and resistant *M. bovis* BCG Russia is shown in Supplementary Fig. 10. The non-normalized growth rate as a function of time in response to drugs are presented in Supplementary Fig. 11.

## Heteroresistance

Heteroresistant infection poses a diagnostic challenge for TB and is linked with suboptimal treatment outcomes[22]. Although our study focuses on rapid pDST, the assay's high throughput and spatial-temporal resolution provide an advantage in detecting hetero-resistance. To test if we could detect 1% resistant bacteria in an otherwise susceptible population, we mixed *M. bovis* BCG Russia WT (susceptible) and one of the resistant strains (*M. bovis* BCG Russia STR^R (RpsL K43R), *M. bovis* BCG Russia INH^R (*katG* delAA428), or *M. bovis* BCG Russia GyrA D94G (FQ^R)) in a 99:1 (v/v) ratio. The resulting mixture was then loaded onto the microfluidic chip, and the assay protocol for *M. bovis* BCG, extended to 48 h, was initiated. STR (Supplementary Fig. 12), INH (Supplementary Fig. 13), and LFX (Supplementary Fig. 14) were introduced into the treatment side at the critical concentrations[61], or at the MIC$_{90}$ for the FQ^R resistant strain (4 mg/L)[60]

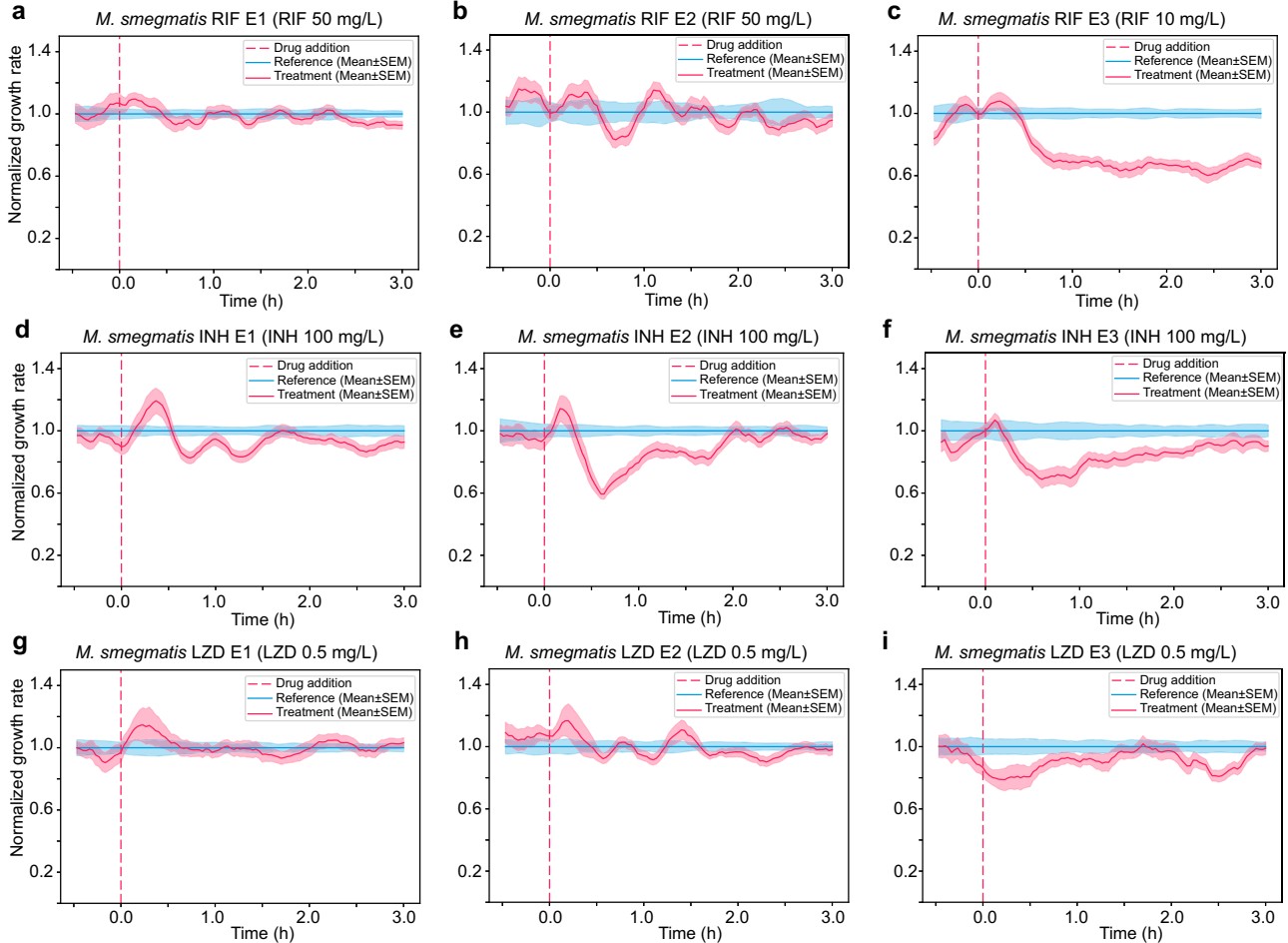

**Fig. 4 | Fast detection of resistant strains.** pDST profiles of lab-evolved resistant strains derived from *M. smegmatis* NCTC 8159 including (**a**) *M. smegmatis* RIF E1, (**b**) *M. smegmatis* RIF E2, (**c**) *M. smegmatis* RIF E3 in rifampicin treatment, (**d**) *M. smegmatis* INH E1, (**e**) *M. smegmatis* INH E2, (**f**) *M. smegmatis* INH E3 in isoniazid treatment, and (**g**) *M. smegmatis* LZD E1, (**h**) *M. smegmatis* LZD E2, (**i**) *M. smegmatis* LZD E3 in linezolid treatment. Data in each graph is from one biological replication. Multiple biological replication data are shown in Supplementary Fig. 8.

(Supplementary Fig. 15). In order to identify microchambers that may contain resistant cells we plot the area growth as a function of time from individual microchambers and compared plots from the reference and treatment (Supplementary Figs. 12a, b, 13a, b, 14a, b, and 15a, b). Individual microchambers that display growth patterns similar to those from the untreated reference were selected as candidates for containing resistant cells (blue boxes in Supplementary Figs. 12b, 13b, 14b, and 15b). We next inspected the movies corresponding to the candidate microchambers manually. In all measurements we could in this way identify individual resistant cells at the detection limit of 1%, (Supplementary Fig. 12e and Supplementary videos 4 and 5, Supplementary Figs. 13e, 14e, and 15e). Overall, the growth profiles of the reference and treatment obtained at high throughput and spatial-temporal resolution could be used to detect heteroresistant infections.

## Discussion

The World Health Organization recommends drug susceptibility testing (DST) for samples from all individuals exhibiting clinical symptoms of active TB, even though existing rapid methods to detect drug resistances (except for rifampicin) are inadequate to achieve this objective[2,21,63,65]. Rapid and accurate results from DST of clinical TB samples are vital to avoid misprescribing ineffective or toxic additional drugs and control the spread of resistant variants. Furthermore, the optimal treatment of each TB patient can be evaluated and, if necessary, altered more rapidly, shortening treatment regimens[18]. Culture-based phenotypic DST (pDST) methods for anti-TB drugs are reliable and reproducible, yet time-consuming, demanding sophisticated lab setups, trained personnel, and rigorous quality control[63]. The pDST presented in this study for slow-growing mycobacteria is inspired by the fast antibiotic susceptibility test (fASTest) developed in our laboratory for UTI pathogens[29]. A clinical test by Sysmex Astrego AB based on this method recently won the Longitude Prize on AMR, demonstrating that it is possible to make related methods clinically useful. In the present study, we modified the chip to accommodate mycobacterial cells and robustly analyzed the image data using a deep neural network.

In the current workflow, we would start a diagnostic assay after a culture-positive TB diagnosis. This avoids the possible challenges of sample prep from sputum and means that we could load relatively high concentrations of exponentially growing bacteria. We currently use bacteria concentrations as low as $5 \times 10^4$ CFU/mL, which could be pushed down further if needed. Another aspect of working with a culture-positive sample is that we would only replace the current two-week pDST, thus not radically changing the TB diagnosis workflow. By applying fast pDST, significant deviations in growth rates between the treatment and reference (control) bacterial populations at MICs of INH, EMB, and LZD are observable within 3 h for slow-growing tuberculous mycobacteria, with a generation time of 18 to 24 h. Growth impact is seen after -1 h for the fast-growing *M. smegmatis*. We applied the rapid pDST on the panel of susceptible and resistant *M. bovis* BCG Russia using

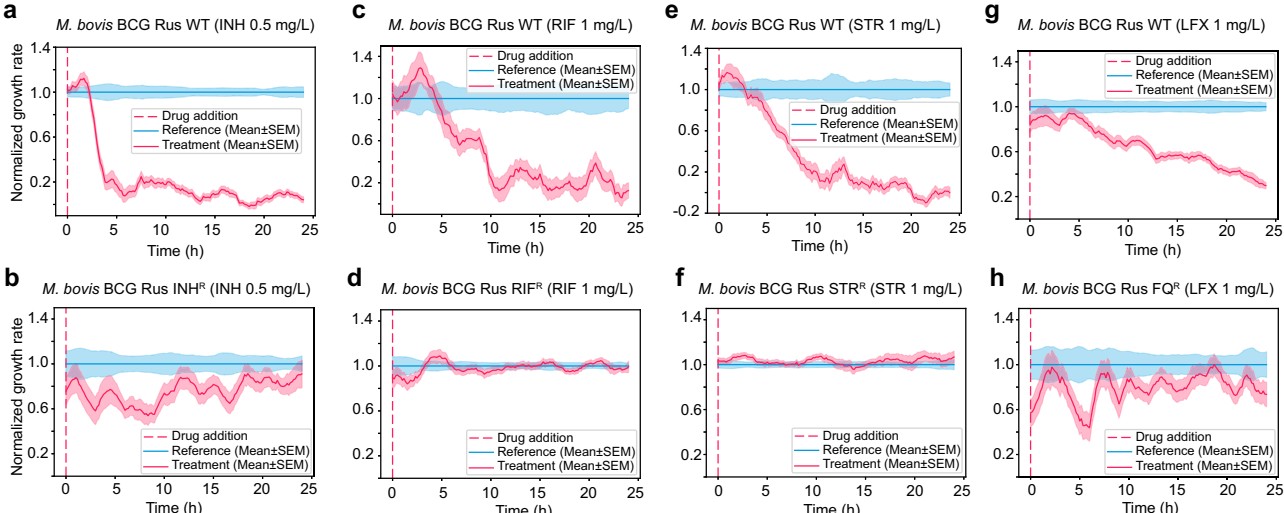

**Fig. 5 | pDST of susceptible and resistant *M. bovis* BCG Russia at critical concentrations (CCs).** pDST profiles of *M. bovis* BCG Russia WT (**a**) and *M. bovis* BCG Russia INH^R (**b**) in INH 0.5 mg/L; *M. bovis* BCG Russia WT (**c**) and *M. bovis* BCG Russia RIF^R (**d**) in RIF 1 mg/L; *M. bovis* BCG Russia WT (**e**) and *M. bovis* BCG Russia STR^R (**f**) in STR 1 mg/L; and *M. bovis* BCG Russia WT (**g**) and *M. bovis* BCG Russia FQ^R (**h**) in LFX 1 mg/L. Data in each graph is from one biological replication. Multiple biological replication data are shown in Supplementary Fig. 10.

critical concentrations (CCs) for INH, RIF, STR, and LFX used in treating TB and drug-resistant TB. The growth deviations between the susceptible and corresponding RIF-, STR-, and FQ-resistant strains were observed 6 h after the drug addition (Fig. 5c–h). INH remarkably and quickly influenced the growth of the treatment population. Still, INH-treated cells required a longer time (12 h) to recover their growth rate to the reference's level (Fig. 5a and b). This phenomenon was also observed in susceptible and resistant *M. smegmatis*.

We also demonstrated the potential for detecting heteroresistance at the detection limit of 1% using our pDST, attributed to its capability to monitor the growth at a high throughput and spatial-temporal resolution (Supplementary Figs. 12–15). Overall, the data from our work presents a rapid pDST for fast- and slow-growing mycobacteria with the potential for heteroresistant detection.

Our pDST is designed to help reduce the diagnostic TAT, enabling faster DST and potentially supporting more effective drug response monitoring, which would benefit patients. Doctors could choose different drug combinations or adjust the dose accordingly. Moreover, the short assay TAT minimizes the acquired resistance developed during the susceptibility testing[27].

Bedaquiline (BDQ) is a lipophilic drug and one of the frontline treatments for MDR-TB. BDQ showed a strong impact on the growth and morphology of the mycobacterial cells in our tests, but gradually lost the impact over time when we used the same medical tubing (Tygon® ND 100–80) as for other drugs. However, rapid and persistent inhibition of *M. smegmatis* growth in response to BDQ was eventually observed using FEB tubing. Overall, one should consider using more chemically inert and highly non-stick tubing such as fluoropolymer (FEB) tubing when working with BDQ in microfluidic setups.

Fast and sensitive methods for TB drug susceptibility testing using direct microscopic observation of broth cultures have been proposed before. However, the median TAT for these tests was 5.4 to 9.5 days[42,66–68]. Our method calculates an accurate growth rate for thousands of cells in multiple microchambers, allowing us to reach statistical significance for growth-rate differences quicker than the corresponding bulk measurement. This work presents the proof-of-principle of a rapid pDST for mycobacteria, particularly the very slow-growing species. The method could potentially integrate with upstream processes for the direct-from-sputum-sample culture,

species detection using trehalose dyes, and pDST in a single run. In the current setup, we implemented the test using a research microscope, followed by subsequent processing of the acquired imaging data. We also used *M. bovis* BCG as a slow-growing tuberculous mycobacterial model because it requires a lower biosafety level. To take the system closer to the clinic, fluid automation in a closed system should be implemented and tested for highly virulent and resistant *M. tuberculosis* from clinical samples.

## Methods

### Microfluidic chip design and fabrication

The microfluidic chip is composed of a molded silicone elastomer layer (polydimethylsiloxane (PDMS); Sylgard 184) covalently bonded with a borosilicate cover glass (thickness no. 1.5, 22 × 40 mm, VWR). The structure of the chip is shown in Supplementary Fig. 1. Details of chip design and features are described in the Supplementary information. The numbering of ports for the microfluidic chip is illustrated in Supplementary Fig. 1. Tubing (VWR, TYGON VERNAAD04103) was connected to the chip via a metal tubing connector. We used ports 5.1, 5.2, and 6.0 to maintain back-channel pressure, port 2.0 to load cells, and ports 2.1 and 2.2 to supply growth media with and without drugs. Flow control was regulated by the OB1-Mk3 regulator (Elveflow).

### Mycobacterial strains and antibiotics

*Mycobacterium tuberculosis* variants *bovis* BCG (ATCC 35734, BCG Pasteur) and *Mycobacterium smegmatis* (mc² 155 and NCTC 8159) were used as models for tuberculous and nontuberculous pathogens. *M. smegmatis* mc² 155 WT was kindly provided by Dr. Leif Kirsebom. *M. smegmatis* NCTC 8159 WT and laboratory-evolved resistant strains [RIF E1 (*rpoB, cypX*), RIF E2 (*rpoB, sugC*), RIF E3 (*rpoB*), INH E1 (*inhA, katG*), INH E2 (*inhA*), INH E3 (*inhA, katG, Rv2042c*), LZD E1 (*oxiR, ugpC, Rv1184c*), LZD E2 (*oxiR, rshA*), and LZD E3 (*oxiR*)] were kindly given by Dr. Tomoya Maeda[59] - mutated genes in the parentheses are ortholog in *M. tuberculosis* H37Rv strain. *M. bovis* BCG Russia WT parental strain, *M. bovis* BCG Russia RIF^R (RpoB S531L), *M. bovis* BCG Russia FQ^R (GyrA D94G), *M. bovis* BCG Russia STR^R (RpsL K43R) were kindly provided by Dr. Marcel Behr[60]. *M. bovis* BCG Russia INH^R (*katG* delAA428) was obtained from the Belgian Coordinated Collections of Microorganisms (CT2020-01217) deposited by Dr. Marcel Behr's laboratory[60]. All *M. bovis* BCG Russia strains harbored the pNIT plasmid containing Kan-R

cassette. Antibiotics rifampicin (RIF, R3501), isoniazid (INH, I3377), ethambutol (EMB, E4630), linezolid (LZD, PHR1885), and streptomycin (STR, S1277) were purchased from Sigma-Aldrich. Bedaquiline (BDQ, HY-14881) and levofloxacin (LFX, HY-B0330) were purchased from MedChemExpress. Stock solutions were prepared by dissolving active agents in DMSO or Milli-Q water (streptomycin) at 10,240 mg/L and stored at −20 °C. Bedaquiline was dissolved in DMSO in combination with ultrasonic and heating to 60 °C as instructed by the manufacturer. FEB tubing (1/16" OD x .020" ID Natural 50 ft) was purchased from Index Health & Science.

## Media and cultural conditions

We used Middlebrook 7H9 (Sigma-Alrich M0178) liquid medium as growth medium (GM) supplied with 10% Acid-Dextrose-Catalase (ADC) solution, 0.005% glycerol (w/v; Sigma-Aldrich G5516), 0.0005% Tween 80 (w/v; Sigma-Aldrich S6760), and 0.17% Pluronic F-108 (w/v; Sigma-Aldrich 542342). ADC solution contains 8.5 g NaCl (Sigma-Aldrich S3014), 50 g bovine serum albumin (Sigma-Aldrich A2153), 20 g dextrose (Sigma-Aldrich D9434), 0.03 g catalase (Sigma-Aldrich C9322) per liter. *M. bovis* BCG from glycerol stocks was streaked on egg-based Löwenstein-Jensen (LJ, Sigma-Aldrich 63237-500G-F) solid plates. *M. smegmatis* was grown on LB agar plates. For off-chip culture, overnight growth was prepared by inoculating three colonies into 4 mL GM in round-bottom clear polystyrene culture tubes (VWR, 734-0435) and incubated at 37 °C with 200 rpm shaking for 18 h and 6 days for *M. smegmatis* and *M. bovis* BCG respectively.

## MIC determination for *M. bovis* BCG pasteur

The MIC measurements for slow-growing *M. bovis* BCG were determined using broth microdilution assay (BMDA) following the guidelines from EUCAST[52]. The guideline was for *M. tuberculosis* but adapted for *M. bovis* BCG. We used 96-well flat-bottom-shaped polystyrene plates (Sigma-Aldrich, Costar 3370). The plates with broth and drugs were prepared and used immediately. The broth was the growth media (GM) for both off-chip and on-chip experiments. The stock concentration of drugs was 10,240 mg/L. A 4X working concentration solution (32 mg/L) was two-step diluted in GM from an aliquot of the stock solution. 0.1 mL of GM was added to all wells except for the peripheral wells, which would be filled with sterile distilled water to prevent desiccation during incubation. Subsequently, 0.1 mL of the 4X working concentration solution was added to the wells corresponding to the highest concentration of each drug - the testing concentration range was 0.06–8 mg/L; No drug was added to the bacteria negative control and growth control (GC) wells. A multichannel pipette was used to make 1:2 dilutions by adding 0.1 mL of the antibiotic solution present in the highest concentration row to the following row and discarding the last 0.1 mL of the last row/wells. The second step is the inoculation of culture from the stationary phase of *M. bovis* BCG. The turbidity of the culture was adjusted to McFarland 0.5. Further dilutions of two concentrations at 1:100 and 1:10000 in GM broth, which corresponded to 100% and 1% "growth control" (GC100% and GC1%), were conducted by tenfold dilution steps. A volume of 0.1 mL of GC100% was added to the wells containing drugs that were prepared in the first step. GC100% and GC1% were also added to designated wells on the plate. The third step is the incubation and MIC determination. After inoculation, a breathable membrane (Sigma-Aldrich, Breathe-Easy® Z380059) was used to seal the plate to avoid drying of the cultures but still kept the air going through, and incubated at 37 °C ± 1 °C under normal shaking conditions in Tecan (Sunrise™, Tecan Nordic AB) incubator. Optical density values were read every 1 h. The negative control should show no growth for the test to be valid. The GC1% should show visible growth and be slower than GC100%. MIC was determined as the lowest concentration of the drug where no visible growth was observed. If there was still insufficient growth of the GC1% after 14 days, incubate until a maximum of 21 days.

## MIC determination for *M. smegmatis*

MIC values for *M. smegmatis* were determined using the microplate-based Alamar Blue assay (MABA)[50,51]. In short, wells on 96-well microtitre plates were filled with 50 µL of growth medium (GM). Double the required antibiotic concentration was prepared and 100 µL volumes were added to the first column. This was serially diluted to half the concentration by mixing with media only in the subsequent wells till the last second column. The last column was a control without antibiotics. The *M. smegmatis* strains were grown in replicates in GM to an optical density (OD 600 nm) of 0.6, diluted to McFarland 0.5, and then further diluted 100-fold. 50 µL of the diluted culture was added to each well such that the final antibiotic concentration in the first well came down to the desired concentration. The plate was sealed with a breathable membrane (Sigma-Aldrich, Breathe-Easy® Z380059) and then incubated under normal shaking conditions in Tecan (Sunrise™, Tecan Nordic) incubator at 37 °C for 40 h. After 40 h of incubation, 30 µL of resazurin dye (Sigma-Aldrich R7017; filter sterilized, 0.2 mg/mL concentration) was added to each well and incubated for 6 h under shaking and then imaged. MICs were determined as the values of the first well showing no growth as indicated by resazurin dye staining.

## Microfluidic experiments

Mycobacteria cells were loaded after diluting 1:100 of a McFarland 0.5 culture, ~from $5 \times 10^4$ to $5 \times 10^5$ CFU/mL[52]. We used a Nikon Ti-E and Nikon Ti2-E inverted microscope equipped with CFI Plan Apochromat DM Lambda 100X oil immersion objectives (Nikon) for imaging *M. bovis* BCG and *M. smegmatis*, respectively. Images of *M. bovis* BCG were captured by the Imaging Source (DMK38UX304) camera and *M. smegmatis* by Sona 4.2B-11 (Andor). The microscopes were controlled by Micro-Manager[69] and in-house built plugins[29,30,70]. The temperature of the microscope stage and the microfluidic chip was maintained at 37 °C using a climate enclosure (Oklab). Because of the growth rate differences between *M. bovis* BCG and *M. smegmatis*, the measurement timeline of 3 + 24 h and 1 + 3 h were applied for *M. bovis* BCG and *M. smegmatis*, respectively. The former parts were the duration without the addition of drugs on both rows (reference and treatment), and the latter part was the duration of drugs added to the media on the row of treatment. The spatial growth indicated by the total cell area in a microchamber was captured throughout the measurement with an interval time of 10 min and 2 min for *M. bovis* BCG and *M. smegmatis*, respectively. About 30–50 microchambers of reference and treatment were monitored simultaneously. The instantaneous growth rates were determined with a sliding window time of 3 hours for *M. bovis* BCG and 30 min for *M. smegmatis*.

For heteroresistance detection experiments, we conducted experiments using the limit of detection threshold of 1% resistant bacteria, *i.e.*, the value obtained for rifampicin DST using the 'gold standard' phenotypic assay (the agar proportion method)[22]. After adjusting the culture turbidity to McFarland 0.5, we mixed *M. bovis* BCG Russia WT and one of the resistant strains (*M. bovis* BCG Russia STR^R (RpsL K43R), *M. bovis* BCG Russia INH^R (katG delAA428), or *M. bovis* BCG Russia GyrA D94G (FQ^R)) at the ratio of 99:1 (v/v). We then loaded the mixture onto the microfluidic chip and started the assay protocol for *M. bovis* BCG with an increased measurement time of 48 hours (3 + 45 hours). The medium on the treatment side was added with STR, INH, and LFX at critical concentrations for *M. tuberculosis*[61] or an MIC$_{90}$ of the FQ^R resistant strain (4 mg/L)[60].

## Image processing

The image data were pre-processed for correcting stage shifts and cropping microchambers as regions of interest using an in-house algorithm developed in MATLAB[29]. Empty microchambers were discarded in further analysis. Cropped phase-contrast images of mycobacterial cells in microchambers were segmented using an image-

segmentation algorithm called Omnipose[44]. We trained new models for the image data of mycobacteria. The training dataset was created with 120 images of manually labeled ground-truth of mycobacterial cells in various experimental conditions including *M. bovis* BCG and *M. smegmatis* on agarose pads and in microchambers, and subjected to antibiotic treatment. Manual labeling ground truth for the training dataset was based on the previous work[30] using LabelsToROI tools and custom scripts. The membrane of cells in high-density microchambers was stained using 3HC-3-Tre to assist in the labeling[48]. Four models were trained from scratch, and two models were fine-tuned from the default Omnipose model for bacterial phase contrast images. We show the training parameters in the Supplementary Table 1. The segmentation performance of new models was quantitatively evaluated by comparing their average Jaccard Index as a function of intersection over union (IoU) threshold on the same dataset.

### Data analysis

The calculation of the growth rate of cells in individual microchambers was completed, as described previously[29,30], by applying a sliding window of data points (area) and fitting an exponential function: $y = ae^{bt}$, where $y$ is the total area of segmented cells in a microchamber, $a$ is the constant, and $b$ is the growth rate. Sliding windows of *M. bovis* BCG and *M. smegmatis* were 3 h and 30 min, respectively. The standard error of the mean (SEM) takes the growth rate standard deviation (SD) and divides it by the square root of the total microchamber number. The growth rate normalization was calculated by dividing the treatment population's mean growth rate by the reference population's mean growth rate at each time interval. The SEM values for both the reference and treatment were normalized by dividing each SEM value at different time intervals by the mean growth rate of the reference population, we refer to this parameter as normalized SEM. The separation time of the treatment population from the reference population was based on the time at the separation of the normalized SEM values between the two populations in the growth rate normalization. For long-term experiments with *M. bovis* BCG, positions losing focus would create outliers in the area growth curve. We removed the outliers before calculating the growth rate by replacing measurements that deviated more than 5% from the curve fitted in the sliding window with the fitted value. After one round of outlier replacement based on the initial curve fitting, the remaining points were refitted to obtain the reported growth rate.

### Statistics and reproducibility

The study is not randomized. The reference and treatment were measured on the same microfluidic chip in the same setup and segmentation model for each measurement. Empty microchambers and microchambers with segmentation errors were not included in the calculation of the growth rate. All pDST profiles display normalized growth rate ± normalized SEM. At least two experimental replicates were conducted for each condition. We carried out one microscopic measurement for *M. bovis* BGC and *M. smegmatis* with multiple positions in each experimental condition to obtain the training dataset for the image segmentation. Representative images are shown in Fig. 2a and Fig. 2d–i.

### Reporting summary

Further information on research design is available in the Nature Portfolio Reporting Summary linked to this article.

## Data availability

All experimental data generated in this study have been deposited in the SciLifeLab Data Repository[71] at https://doi.org/10.17044/scilifelab.26927146. Source data are provided with this paper.

## Code availability

All code used in this paper is available in the SciLifeLab Data Repository[71] at https://doi.org/10.17044/scilifelab.26927146.

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

## Acknowledgements

We acknowledge funding from the Swedish Foundation for Strategic Research (SSF ARC19-0016), Knut and Alice Wallenberg Foundation KAW (2023.0531), and Novo Nordisk (0083419). All fundings were awarded to J.E. We thank Irmeli Barkefors for her helpful comments on the manuscript, and Dr. Tomoya Maeda and Dr. Leif Kirsebom for providing *M. smegmatis* strains. We thank Dr. Marcel Behr and Dr. Ori Solomon for providing *M. bovis* BCG Russia strains.

## Author contributions

J.E. conceived the pDST method for mycobacteria and supervised the project. B.M.T. performed pDST of *M. bovis* BCG and all data analysis. J.L. performed pDST of *M. smegmatis* NCTC 8159 strains. A.G. performed pDST of *M. smegmatis* mc$^2$ 155. P.K. trained mycobact_1 to mycobact_4 models and wrote scripts for model evaluation. J.E. and B.M.T. wrote the paper with input from all authors.

## Funding

## Competing interests

J.E. has patented the method (US10,041,104B) and founded Astrego Diagnostics, but he has no current association with that company. No current company is associated with this work, but there may be in the future. All other authors declare no competing interests.
