## [Transparent Peer Review file · Nature Communications]

Phenotypic drug susceptibility testing for *Mycobacterium tuberculosis* variant *bovis* BCG in 12 hours

Corresponding Author: Professor Johan Elf

Version 0:

Reviewer comments:

Reviewer #1

(Remarks to the Author)

Summary

The authors present early developmental work on a novel drug susceptibility test that could significantly reduce the time to resistance detection using phenotypic methods for *M.tuberculosis*. The combination of microchambers, high-end microscopy imaging and deep neural network-based analytic methods holds much promise. The proof of principle demonstrates the potential to work, though it used proxy organisms, and additional work to test the performance of *M. tuberculosis* and its different lineages would be an important next step. The growth curves show distinct patterns between susceptible and resistant strains and in addition provides time-series observations which could potentially provide insights on persistence and tolerance which have been a topic of interest for many years and this could provide direct biological observations. Overall the study is appropriate and the methodology and results support the conclusions. There are a few issues that need to be addressed:

Major comments

Main text

1. Page 2 Line 14: suggested addition of "particularly XDR-TB" in the sentence: "Treating drug-resistant TB, particularly XDR-TB," ... as the treatments for RR-TB are now 6 to 9 months with improved outcomes though for XDR this is not the case.
2. Page 2 Line 33-34: the statement is inaccurate "cannot detect novel 33 resistance mutations" – WGS is used specifically for this purpose. Note limitations for molecular test is the inability to detect resistance to the new drug e.g bedaquiline, delamanid etc.
3. Page 2 Line 34-35: also inaccurate: "not that it is susceptible." The UK uses WGS to predict susceptibility for first line drugs while the recently WHO recommended tNGS systems report susceptibility.
4. It is a pity that other important drugs such as fluoroquinolones or bedaquiline were not included. It would be important to understand the reason – is there particular issues with such drugs and this assay or was it for convenience?
5. Page 9 Line 14-15: Statements like: "10x MIC of RIF (0.6mg/L)" are unclear. What was the RIF MIC of the strain? 0.6mg/L is that what was used? Was the assay chambers with this strain tested at concentrations above and below this value and what were they? What was the cut-off criteria used to call a strain susceptible or resistant by the reference method.
6. In the discussion, it would be useful to discuss the potential of the assay for detecting heteroresistance since the assay is analysing a per-cell level. Heteroresistance is emerging as an important issue for fluoroquinolones and bedaquiline – two key second line drugs.
7. Page 14 Line 10-12: It is not clear – a 0.5 McFarland was used and dilutions of 1:100 was done which would equate to a 1% not a 100%. Is there a step missing in the description that makes this a neat 100% concentration?
8. Page 14 Line 4: More details are required on the 4x working solution. What was the concentration of the working solution?

Minor comments

Abstract

1. Page 1 Line 18-19: Not clear what response time at MIC means
2. Page 1 Line 12-13: in addition to the organism, mention antibiotics evaluated
3. Page 1 Line 20: results for both organisms stated for susceptible strains but here for resistance only *M.smegmatis* results are presented. Add missing information
4. Page 2 Line 5: mentioning of latent infection is not relevant in a DR-TB paper and sentence can be removed
5. Page 2 Line 20: change "are enforced" to "be used" as I do not think people or programs can be forced to do anything.

6. Page 12 Line 5 – the statement is not totally correct. The detection of rifampicin resistance using molecular tests is highly accurate while other tests may perform less well. The statement “despite the existing tools being” should be restated and balanced to state for some drugs testing accuracy is inadequate.
7. Page 12 Line 37; The statement is too string and would suggest adding in “potential”

Reviewer #2

(Remarks to the Author)

The manuscript by Elf and colleagues presents a promising image-based method for rapid phenotypic drug susceptibility testing (pDST) with the tuberculosis models *Mycobacterium smegmatis* and *Mycobacterium bovis* BCG. By using a custom-made microfluidic chip and a deep neural network-based segmentation algorithm, the authors claim that drug resistance in these model microorganisms can be detected in a significantly reduced timeframe compared to conventional methods used in the clinic. The proposed methodology has the potential to decrease the turnaround time of pDST for tuberculosis, thereby supporting quicker and more accurate treatment.

Tuberculosis is one of the main causes of antibiotic resistance and poses severe diagnostic limitations, which significantly impact patient management and disease transmission. While the manuscript strives to address these important issues, the technical robustness of the approach would benefit from further development, and additional evidence is needed to convincingly demonstrate its diagnostic potential. The use of microfluidics and single cell segmentation for antimicrobial testing has been extensively applied in bacterial research and is not novel for mycobacterial species, including *Mycobacterium tuberculosis*, as reported in recent publications (PMID: 36379978; PMID: 38755132). Despite what can be inferred from the title, most of the data presented in this manuscript derives from the non-pathogenic surrogate *M. smegmatis*, which has limited relevance to clinical tuberculosis diagnostics and raises questions regarding the translational value of the proposed method for *M. tuberculosis*. More specific major concerns are indicated below.

1. The technical novelty of the manuscript is not completely clear, as similar microfluidic chips were previously developed by the team but applied to different bacterial species. Furthermore, although the authors mention that detailed descriptions of the chip are available in the Supplementary Information, these details are lacking. Without explicit technical information on the design, fabrication, and use of the microfluidic chip, independent replication or further application by other scientists will not be possible.
2. Novelty is also questionable with respect to the image segmentation algorithm that is derived from the popular Omnipose. The authors also fail to acknowledge prior applications of Omnipose in segmenting mycobacterial cells.
3. Based on Figure 2a, it is difficult to understand the actual utility of the fluorescent membrane dye to help annotate ground truth images, as bacteria overlap in many areas of the chip, leading to an additive fluorescence signal. This, in addition to the fact that mycobacteria tend to move within the chip chamber, makes accurate segmentation virtually impossible. As a result, the segmentation performance of the best Mycobact_2* model appears to be suboptimal, given the limited ability of the model to distinguish single cells in high-density cell groups. Indeed, by inspecting representative masks, several ROIs contain clusters of bacteria rather than individual cells, undermining the precision of the proposed single-cell pDST. Overall, this might explain why data are presented as averaged values, overlooking information on single cells, and raising questions about the practical advantages of the method. Incidentally, Table 1 is not accessible to non-specialists.
4. What is the advantage of showing different biological replicates split between main and supplementary figures? This makes the findings less accessible to the reader.
5. The authors measured changes in the population growth rate of drug-susceptible *M. smegmatis* or *M. bovis* BCG and in *M. smegmatis* strains carrying specific drug target mutations. No other clinically relevant parameter has been inferred from the analyzed datasets, making it difficult to assess the advantage of the proposed methodology over existing methods. To prove the translational relevance of this approach, validation with *M. tuberculosis* and clinical isolates and testing different drug concentrations would be essential. Additionally, the proposed methodology is unlikely to be able to detect more complex resistance profiles, such as heteroresistance and phenotypic tolerance, thus having limited impact on tuberculosis diagnostics.

In its current form, I do not consider this manuscript suitable for publication in *Nature Communications*. To improve the manuscript and increase its translational relevance, the authors should demonstrate significant advances in both the microfluidic chip and analytical pipeline, include comprehensive details of the methods to support reproducibility, validate the findings with clinically relevant *M. tuberculosis* strains and drug concentrations, and carry out more extensive analysis of the data in relation to PK-PD parameters.

Reviewer #3

(Remarks to the Author)

One of the challenges in managing the burden of TB disease is the long duration of time needed for drug susceptibility testing (DST). Typically, DST is performed by phenotypic (liquid culture-BACTEC or solid media-based culture) or genotypic methods (nucleic acid amplification tests or whole genome sequencing). These have turnaround times ranging from 90

minutes (GenXPert) to 14 days (BACTEC MGIT).

In the current manuscript, the authors have implemented the combination of microfluidics, microscopy and neural network-based image analysis for phenotypic DST of mycobacterial populations. This approach has been used previously by their group for drug sensitivity testing of uropathogenic *E. coli*. Testing was carried out using fast growing (*M. smegmatis*) and slow growing (*M. bovis* BCG) mycobacteria. Against susceptible strains, growth rate differences against anti-TB drugs were detected within 3h in case of *M. bovis* BCG and under 1h in case of *M. smegmatis*. When evaluating the response of candidate resistant strains of *M. smegmatis*, the detection window increased to 3h.

The study is well-designed and executed. The text and data presentation are very clear and flows well. Overall, while the study is not entirely novel or the approach revolutionary, it does represent a significant advance for *M. tuberculosis* DST testing especially with regards to the image analysis. The study would have been strengthened significantly if the testing was carried out using *M. tuberculosis*, especially clinical isolates in this setup and using antibiotic breakpoint concentrations exposures.

Specific comments:

- The authors posit (Pg 2, Ln 31-35) that genotypic DST approaches such as whole genome sequencing are expensive, unfit for point-of-care testing and inaccessible to low-resource areas. These arguments can also be said of the methodology proposed by them. In the current form, it is difficult to envisage that this approach will be widely used in low-resource settings as it would need specialized equipment as well as image analysis expertise to process the data.

- Missing reference Wang et al., (2021), Pg 3 Ln 27

- The data shown in Figs 1d-1i measures the total cell area within the microchambers. So the use of the term "single-cell growth" is misleading and should be modified.

- It is not clear from the current dataset provided how the choice is made for defining the concentration of antibiotic for measuring resistance. For example, 1X MIC was used in case of INH against BCG whereas 10X MIC was used for Rifampicin. Will this have to be determined empirically or is this dependent upon the mode of action of antibiotics? Also, how does the technique work when analyzing a heteroresistant isolate.

- The workflow in the current form addresses the drug susceptibility testing and speeds it up but is still reliant on the culture of the inoculum to feed into the microfluidic chip. Can this approach be directly used on sputum samples itself, combining it with bacterial species identification as well, for example using the mycobacteria-specific trehalose dyes.

- What is the nature of the mutation in RIF E3? The MIC of the strain is reported to be 50 mg/L, yet it seems to be inhibited by exposure to RIF 10 mg/L.

Version 1:

Reviewer comments:

Reviewer #1

(Remarks to the Author)
No further comments

Reviewer #2

(Remarks to the Author)
The revised version of the manuscript by Elf and colleagues addresses my previous concerns, and I consider the manuscript suitable for publication in its current form.

Reviewer #3

(Remarks to the Author)
The authors have revised the manuscript incorporating the feedback from the previous reviews and have also included additional data on two other antibiotics, drug resistant strains and monitoring of heteroresistance. These revisions have strengthened the manuscript significantly and I have no other major comments.

- Supplementary information File, Page 3 Ln 17 "We used ports 2.1 and 2.2 to supply media without and without antibiotics." should be "We used ports 2.1 and 2.2 to supply media with and without antibiotics."

RESPONSE TO REVIEWER COMMENTS

Reviewer #1 (Remarks to the Author):

Summary

The authors present early developmental work on a novel drug susceptibility test that could significantly reduce the time to resistance detection using phenotypic methods for M.tuberculosis. The combination of microchambers, high-end microscopy imaging and deep neural network-based analytic methods holds much promise. The proof of principle demonstrates the potential to work, though it used proxy organisms, and additional work to test the performance of M. tuberculosis and its different lineages would be an important next step. The growth curves show distinct patterns between susceptible and resistant strains and in addition provides time-series observations which could potentially provide insights on persistence and tolerance which have been a topic of interest for many years and this could provide direct biological observations. Overall the study is appropriate and the methodology and results support the conclusions. There are a few issues that need to be addressed:

Reply: we thank the reviewer for their kind words and recognition of our work's innovation. In the paragraphs below, we addressed the reviewer's comments point-by-point.

Major comments

Main text

1. Page 2 Line 14: suggested addition of "particularly XDR-TB" in the sentence: "Treating drug-resistant TB, particularly XDR-TB," ... as the treatments for RR-TB are now 6 to 9 months with improved outcomes though for XDR this is not the case.

Reply: we agree with reviewer 1 on this point and added "particularly XDR-TB" to the sentence. (Page 2, Line 14)

2. Page 2 Line 33-34: the statement is inaccurate "cannot detect novel 33 resistance mutations" – WGS is used specifically for this purpose. Note limitations for molecular test is the inability to detect resistance to the new drug e.g. bedaquiline, delamanid etc.

Reply: we modified the text to clarify that the WGS methodology is incapable of detecting resistance to the new drugs e.g. bedaquiline, delamanid etc. (Page 2, Line 34 - 35)

3. Page 2 Line 34-35: also inaccurate: "not that it is susceptible." The UK uses WGS to predict susceptibility for first line drugs while the recently WHO recommended tNGS systems report susceptibility.

Reply: we thank the reviewer for pointing out the inaccuracy. We have modified the sentence to clarify it. (Page 2, Line 35 - 38)

4. It is a pity that other important drugs such as fluoroquinolones or bedaquiline were not included. It would be important to understand the reason – is there particular issues with such drugs and this assay or was it for convenience?

Reply: In the additional measurements, we carried out the pDST on the panel of susceptible and resistant *M. bovis* BCG Russia strains at the critical concentrations (CCs) for *M. tuberculosis* (Page 12, Line 8 - 18; Page 13, Line 1 - 16). Data from these measurements are shown in Figure 5 and Supplementary Fig. 10. In the additional data, we included two antibiotics: streptomycin (STR) and levofloxacin (LFX, a fluoroquinolone used in TB treatment) on the susceptible and resistant strains of *M. bovis* BCG Russia.

We also tested the assay using bedaquiline (BDQ). After significant troubleshooting, we realized that the medical tubing (Tygon® ND 100-80) was unsuitable for BDQ. When we switched to chemically inert fluorinated ethylene propylene (FEP) tubing, we observed the rapid and indefinite growth inhibition of susceptible *M. smegmatis* at 1X MIC (0.05 mg/L) BDQ using FEP tubing (Supplementary Fig. 7f). The manuscript describes more details (Pages 10, Lines 10 - 22; page 11, Lines 1 - 2).

5. Page 9 Line 14-15: Statements like: “10x MIC of RIF (0.6mg/L)” are unclear. What was the RIF MIC of the strain? 0.6mg/L is that what was used? Was the assay chambers with this strain tested at concentrations above and below this value and what were they? What was the cut-off criteria used to call a strain susceptible or resistant by the reference method.

Reply:

In the first version of the manuscript, we used antibiotic concentrations related to the MIC of wildtype and typically not the higher critical concentrations, which would give faster responses but may also impact resistant isolates, which we did not have access to at the time. For example, the critical concentrations (CCs) of RIF, EMB, and LZD in liquid culture by WHO for *M. tuberculosis* are 1 mg/L, 5 mg/L and 1 mg/L, respectively¹⁻³, whereas the MICs are 0.06 mg/L, 2 mg/L, and 0.5 mg/L, respectively (this study).

In the revised version, we had access to susceptible and resistant *M. bovis* BCG Russia strains. We added the results from pDST measurements on those new strains at the CCs for *M. tuberculosis* (Page 12, Line 8 - 18; Page 13, Line 1 - 16).

With respect to the specific question about RIF, we started with 1X MIC at 0.06 mg/L and observed the growth difference after about 8 hours (Supplementary Figure 2b). We then increased the RIF concentration to 10X MIC (0.6 mg/L, still lower than the CC for RIF at 1 mg/L) and observed the growth difference under 3 hours (Figure 3a and Supplementary Figure 2a).

6. In the discussion, it would be useful to discuss the potential of the assay for detecting heteroresistance since the assay is analysing a per-cell level. Heteroresistance is emerging as an important issue for fluoroquinolones and bedaquiline – two key second line drugs.

Reply: The capability of seeing individual growing cells among nongrowing cells provides an advantage in detecting heteroresistance. To test if we could detect 1% resistant bacteria in an

otherwise susceptible population, we mixed *M. bovis* BCG Russia WT (susceptible) and one of the resistant strains (*M. bovis* BCG Russia STR^R (RpsL K43R), *M. bovis* BCG Russia INH^R (*katG* delAA428), or *M. bovis* BCG Russia GyrA D94G (FQ^R)) in a 99:1 (v/v) ratio (Page 13, Line 18 - 28; Page 14, Line 1 - 12). The results showed the potential to detect heteroresistance because the method could track the growth in single microchambers, which allowed us to find the individual resistant cells. A larger study would be needed to determine the limit of detection for heteroresistance and to develop automatic methods for detection.

7. Page 14 Line 10-12: It is not clear – a 0.5 McFarland was used and dilutions of 1:100 was done which would equate to a 1% not a 100%. Is there a step missing in the description that makes this a neat 100% concentration?

Reply: According to EUCAST guidelines for MIC determination of *Mycobacterium tuberculosis* complex isolates using broth microdilution⁴, the final inoculum, which is 100% growth control, is a 10⁵ CFU suspension, obtained from a 10⁻² dilution of a 0.5 McFarland suspension. Reading is done as soon as the 1:100 diluted control (i.e. 10³ CFU/mL suspension, or 1% growth control which is 1:10000 of a 0.5 McFarland suspension) shows visual growth. In other words, dilution 1:100 of 0.5 McFarland is to obtain 100% growth control and further dilution 1:10000 of 0.5 McFarland is to obtain 1% growth control.

8. Page 14 Line 4: More details are required on the 4x working solution. What was the concentration of the working solution?

Reply: We thank the reviewer for pointing out the usage of an unclear phrase. We added details to clarify the sentence (Page 17, Line 4 - 9). The 4X working solution means 4X working concentration solution, defined in the EUCAST guidelines⁴ (see Table 2 in the guidelines), which is 4X higher than the highest concentration in the testing range. For example, if we wanted to test Rifampicin ranging from 0.06 to 8 mg/L, we needed to prepare 4X working concentration solution (32 mg/L) in the growth medium after two times dilutions (1:64 and 1:5) also in the growth medium from the stock concentration of 10240 mg/L. In the protocol, 0.1 mL of the 4X working concentration solution (32 mg/L) was added to the well corresponding to the highest drug concentration (expected to be 8 mg/L) already contained 0.1 mL growth medium making it 16 mg/L. After that, 0.1 mL was taken for 1:2 dilution to the next well. The well was left with 0.1 mL which was then added with 0.1 mL of the inoculum making it finally 8 mg/L.

Minor comments

Abstract

1. Page 1 Line 18-19: Not clear what response time at MIC means

Reply: We have substantially changed this part in the revised manuscript (Page 1, Line 21 - 24)

2. Page 1 Line 12-13: in addition to the organism, mention antibiotics evaluated

Reply: We added the antibiotics used in our study to the abstract. "... - *i.e.* rifampicin (RIF), isoniazid (INH), ethambutol (EMB), linezolid (LZD), streptomycin (STR), bedaquiline (BDQ), and levofloxacin (LFX)..." (Page 1, Line 15 - 17)

3. Page 1 Line 20: results for both organisms stated for susceptible strains but here for resistance only *M. smegmatis* results are presented. Add missing information

Reply: We have added the missing information requested by the reviewer. In the additional measurements, we carried out the pDST on the panel of susceptible and resistant *M. bovis* BCG Russia strains at the critical concentrations (CCs) for *M. tuberculosis* by WHO (Page 12, Line 8 - 18; Page 13, Line 1 - 16). Data from these measurements are shown in Figure 5 and Supplementary Fig. 10.

4. Page 2 Line 5: mentioning of latent infection is not relevant in a DR-TB paper and sentence can be removed

Reply: This is a reasonable point. We omitted the sentence as suggested. (Page 2, Line 5 - 6)

5. Page 2 Line 20: change "are enforced" to "be used" as I do not think people or programs can be forced to do anything.

Reply: We agree with the reviewer and rephrased the sentence. (Page 2, Line 20)

6. Page 12 Line 5 – the statement is not totally correct. The detection of rifampicin resistance using molecular tests is highly accurate while other tests may perform less well. The statement "despite the existing tools being" should be restated and balanced to state for some drugs testing accuracy is inadequate.

Reply: We agree with the reviewer and add more information to justify the stage of rifampicin. "..., even though existing rapid methods to detect drug resistances (except for rifampicin) are inadequate to achieve this objective". (Page 14, Line 15 - 16)

7. Page 12 Line 37; The statement is too strong and would suggest adding in "potential"

Reply: We have modified the statement as the reviewer suggested to make it more measured. "Our pDST is designed to help reduce the diagnostic turnaround time (TAT), enabling faster DST and potentially supporting more effective drug response monitoring, which would benefit patients." (Page 15, Line 13 - 15)

Reviewer #2 (Remarks to the Author):

The manuscript by Elf and colleagues presents a promising image-based method for rapid phenotypic drug susceptibility testing (pDST) with the tuberculosis models *Mycobacterium smegmatis* and *Mycobacterium bovis* BCG. By using a custom-made microfluidic chip and a deep neural network-based segmentation algorithm, the authors claim that drug resistance in these model microorganisms can be detected in a significantly reduced timeframe compared to

conventional methods used in the clinic. The proposed methodology has the potential to decrease the turnaround time of pDST for tuberculosis, thereby supporting quicker and more accurate treatment.

Tuberculosis is one of the main causes of antibiotic resistance and poses severe diagnostic limitations, which significantly impact patient management and disease transmission. While the manuscript strives to address these important issues, the technical robustness of the approach would benefit from further development, and additional evidence is needed to convincingly demonstrate its diagnostic potential. The use of microfluidics and single cell segmentation for antimicrobial testing has been extensively applied in bacterial research and is not novel for mycobacterial species, including *Mycobacterium tuberculosis*, as reported in recent publications (PMID: 36379978; PMID: 38755132). Despite what can be inferred from the title, most of the data presented in this manuscript derives from the non-pathogenic surrogate *M. smegmatis*, which has limited relevance to clinical tuberculosis diagnostics and raises questions regarding the translational value of the proposed method for *M. tuberculosis*. More specific major concerns are indicated below.

Reply: We thank the reviewer for their comments and suggestions. We have addressed each of the reviewer's comments in the following paragraphs.

Comments

1. The technical novelty of the manuscript is not completely clear, as similar microfluidic chips were previously developed by the team but applied to different bacterial species. Furthermore, although the authors mention that detailed descriptions of the chip are available in the Supplementary Information, these details are lacking. Without explicit technical information on the design, fabrication, and use of the microfluidic chip, independent replication or further application by other scientists will not be possible.

Reply: We thank the reviewer for the comments. The technical novelty of our work lies not only in the microfluidic chip or the image segmentation algorithm but in the integration of microfluidics, time-lapse phase contrast microscopy, and advanced image analysis to rapidly perform pDST in mycobacteria, particularly slow-growing species. In the manuscript, we acknowledged some of the prior attempts using microfluidic chips and microscopy to study the physiology and antibiotic response of mycobacteria (Page 3, Line 24 - 34). However, not all of those works aimed to develop a pDST for slow-growing species which is the main focus of our study. We thank the reviewer for explicitly suggesting the references (PMID: 36379978; PMID: 38755132), and we have involved them in our revised manuscript. Our chip's design is different from the chips in the references: (1) Constriction design facilitating the loading, trapping cells, and passing liquid media around the cells; (2) A larger number of microchambers providing the reproducibility in each condition or facilitating the detection of heteroresistance or growth variation; and (3) Cells growing in microchambers without being forced/pressed from the top layer.

As described in our manuscript, the main feature of the chip is a cell trap region that was redesigned to accommodate mycobacteria. As requested by the reviewer, we have added the

supplementary materials and methods to the Supplementary Information where we explicitly described details of the chip design and fabrication as well as the macrofluidic set-up and microfluidic chip wetting before loading cells (Pages 2 - 3, Supplementary Information). We also added Supplementary Fig. 1 including the drawing design of the microfluidic chip displaying the designated ports, filter region with orientations, chip number, chip type/number, ad manifold positions for multiple modes of operations, and microcopy images detailing different regions inside the chip (Page 3, Supplementary Information).

2. Novelty is also questionable with respect to the image segmentation algorithm that is derived from the popular Omnipose. The authors also fail to acknowledge prior applications of Omnipose in segmenting mycobacterial cells.

Reply: As mentioned in the previous answer, the segmentation algorithm is not the novelty presented in our work. We used Omnipose as it provided a robust, high accuracy, and morphologically independent segmentation for elongated cells. We employed the Omnipose algorithm in another study for segmenting mixed bacteria samples⁵. We learnt that it was still suitable for the segmentation of mycobacterial cells once using a model trained on a proper mycobacteria database. Therefore, we carried out the training and evaluation of new segmentation models suitable for the mycobacteria dataset. We thank the reviewer for pointing out the application of Omnipose in segmenting mycobacterial cells in prior work⁶ and we have added missing information in the manuscript (Page 3, Line 31 - 33; Page 6, Line 21 - 22).

3. Based on Figure 2a, it is difficult to understand the actual utility of the fluorescent membrane dye to help annotate ground truth images, as bacteria overlap in many areas of the chip, leading to an additive fluorescence signal. This, in addition to the fact that mycobacteria tend to move within the chip chamber, makes accurate segmentation virtually impossible. As a result, the segmentation performance of the best Mycobact_2* model appears to be suboptimal, given the limited ability of the model to distinguish single cells in high-density cell groups. Indeed, by inspecting representative masks, several ROIs contain clusters of bacteria rather than individual cells, undermining the precision of the proposed single-cell pDST. Overall, this might explain why data are presented as averaged values, overlooking information on single cells, and raising questions about the practical advantages of the method. Incidentally, Table 1 is not accessible to non-specialists.

Reply: We used fluorescent membrane dye to guide the labelling of the contour of mycobacterial cells, particularly in high-density regions. We used the dye to mark the ground truth as much as possible, but at overlapping areas, we could only mark the contour of the blobs. That explains the segmenting limitation of the model at high-density clumps as mentioned by the reviewer. We also discussed this limitation in the manuscript (Page 7, Line 9 - 11). However, in lower cell density regions the performance of newly trained models significantly improved compared to a default model from Omnipose (Fig. 2d and 2e). We used the chip with a thickness of 1 μm to obtain a single layer of cells and precisely quantify the expansion of cells in 2D. Still, mycobacterial cells appeared to squeeze and/or overlap, making it difficult to be single-cell segmented in that condition. With 2D imaging, absolute single-cell segmentation might not be achieved with available segmenting algorithms because of the missing information on the 3rd

dimension. Regardless of the suboptimal segmentation at high-density clumps, the growth rate differences between the reference and treatment populations for different drugs at MIC were rapidly observed. Overall, we sought to achieve an absolute quantification of single-cell growth rate and we could do it if cells were well-separated in the microchambers. Additionally, with a total assay time of 27 hours, severe overlapping of growing cells is unlikely for slow-growing mycobacteria species from dilute samples.

We have modified the parameters in Supplementary Table 1 to make it more accessible to non-specialists (Page 17, Supplementary Information).

4. What is the advantage of showing different biological replicates split between main and supplementary figures? This makes the findings less accessible to the reader.

Reply: We thank the reviewer for the question. We (obviously) observe a smaller range of SEM and smoother curves with increased biological replicates (Supplementary Figures 2, 3, 8, and 10). However, for a diagnostic method, the difference in antibiotic responses between the reference and treatment populations must be observed and confirmed in one biological replication rather than the average over numerous replications.

5. The authors measured changes in the population growth rate of drug-susceptible *M. smegmatis* or *M. bovis* BCG and in *M. smegmatis* strains carrying specific drug target mutations. No other clinically relevant parameter has been inferred from the analyzed datasets, making it difficult to assess the advantage of the proposed methodology over existing methods. To prove the translational relevance of this approach, validation with *M. tuberculosis* and clinical isolates and testing different drug concentrations would be essential.

Additionally, the proposed methodology is unlikely to be able to detect more complex resistance profiles, such as heteroresistance and phenotypic tolerance, thus having limited impact on tuberculosis diagnostics.

Reply: As a proof of concept study, we aimed to reduce the pDST in TB from the range of days to weeks down to the range of hours. To this aim, we used the surrogate strains commonly used in studying drug responses in TB research. We used both fast- and slow-growing models. We understand that the validation of the method would be increased if we could use *M. tuberculosis*, clinical isolates or samples directly from the patient's sputum, but stepping into a BSL3 facility would be a huge hurdle compared to BSL2. However, this study provides a solid foundation for us, and potentially others, to further develop this rapid pDST for use in BSL3 with clinical isolates.

In the revised version, we instead increased the study to a panel of antibiotic-susceptible and -resistant mycobacterial strains with attenuated virulence (*M. bovis* BCG Russia)⁷. The panel was created to provide safe quality control reagents for detecting drug-resistant *Mtb*⁷. The additional measurements were carried out at the critical concentrations (CCs) for *M. tuberculosis* by WHO (Page 12, Line 8 - 18; Page 13, Line 1 - 16). Data from these measurements are shown in Figure 5 and Supplementary Figure 10.

The method could detect heteroresistance and phenotypic tolerance as it can identify a small number of growing cells among many non-growing. We have added experiments for simulated heteroresistant infections detection (see Answer 6, Reviewer 1; and Page 13, Line 18 - 28 plus Page 14, Line 1 - 12). The results suggest that it is possible to detect heteroresistance because the method could track the growth of single microchambers and single cells in microchambers.

In its current form, I do not consider this manuscript suitable for publication in *Nature Communications*. To improve the manuscript and increase its translational relevance, the authors should demonstrate significant advances in both the microfluidic chip and analytical pipeline, include comprehensive details of the methods to support reproducibility, validate the findings with clinically relevant *M. tuberculosis* strains and drug concentrations, and carry out more extensive analysis of the data in relation to PK-PD parameters.

Reply: We have added details of the chip design and fabrication process (see Answer for question 1). We have answered the question regarding *M. tuberculosis* and clinical isolates (see Answer for question 5). We have open-access published our raw data and analysis code. The concept of conducting PK-PD studies using our method is compelling. While our methodology and microfluidic chip provide a robust and versatile tool for such studies, PK-PD analysis lies beyond the scope of this work.

Reviewer #3 (Remarks to the Author):

One of the challenges in managing the burden of TB disease is the long duration of time needed for drug susceptibility testing (DST). Typically, DST is performed by phenotypic (liquid culture-BACTEC or solid media-based culture) or genotypic methods (nucleic acid amplification tests or whole genome sequencing). These have turnaround times ranging from 90 minutes (GenXPert) to 14 days (BACTEC MGIT).

In the current manuscript, the authors have implemented the combination of microfluidics, microscopy and neural network-based image analysis for phenotypic DST of mycobacterial populations. This approach has been used previously by their group for drug sensitivity testing of uropathogenic *E. coli*. Testing was carried out using fast growing (*M. smegmatis*) and slow growing (*M. bovis* BCG) mycobacteria. Against susceptible strains, growth rate differences against anti-TB drugs were detected within 3h in case of *M. bovis* BCG and under 1h in case of *M. smegmatis*. When evaluating the response of candidate resistant strains of *M. smegmatis*, the detection window increased to 3h.

The study is well-designed and executed. The text and data presentation are very clear and flows well. Overall, while the study is not entirely novel or the approach revolutionary, it does represent a significant advance for *M. tuberculosis* DST testing especially with regards to the image analysis. The study would have been strengthened significantly if the testing was carried out using *M. tuberculosis*, especially clinical isolates in this setup and using antibiotic breakpoint concentrations exposures.

Reply: We thank the reviewer for the positive comments and suggestions to strengthen the method. Although we have not yet been able to access the BSL-3 facility to work with pathogenic *M. tuberculosis*, we have access to resistant *M. bovis* BCG, allowing us to conduct additional experiments at critical concentrations. Without resistant isolates, we were concerned that using critical concentrations of antibiotics may also impact resistant isolates.

Specific comments:

- The authors posit (Pg 2, Ln 31-35) that genotypic DST approaches such as whole genome sequencing are expensive, unfit for point-of-care testing and inaccessible to low-resource areas. These arguments can also be said of the methodology proposed by them. In the current form, it is difficult to envisage that this approach will be widely used in low-resource settings as it would need specialized equipment as well as image analysis expertise to process the data.

Reply: We have modified some of the text about the genotypic test and WGS as suggested by reviewer 1 (Page 2, Line 32 - 38), but we still kept the argument about the cost of WGS. At this point, it is hard for us to predict the price of a final product based on the suggested methods. We have, however, previously been able to develop a microfluidic test for rapid AST in urinary tract infections that is now sold by Sysmex corporation for point-of-care use, *i.e.*, the PA-100 system that won the Longitude prize for AMR. Potential for us in the LMIC setting was one of the criteria for winning the prize. The user-friendliness and cost could be similar to a TB test. In the last paragraph of the discussion, we acknowledge that “This work presents the proof-of-principle of a rapid pDST for mycobacteria, particularly the very slow-growing species. In the current setup, we implemented the test using a research microscope followed by subsequent processing of the acquired imaging data.” (Page 15, Line 29 - 34). However, in our experience, the final diagnostic instrument could be much cheaper, user-friendly, and fully automated.

- Missing reference Wang et al., (2021), Pg 3 Ln 27

Reply: We thank the reviewer for pointing out the missing reference. We have added the missing reference. (Page 3, Line 30 - 31)

- The data shown in Figs 1d-1i measures the total cell area within the microchambers. So the use of the term “single-cell growth” is misleading and should be modified.

Reply: In the segmentation section, we acknowledged that “High cell density data (e.g. clumps or cords) were problematic due to overlapping cells and difficulty distinguishing single-cell boundaries. Instead of identifying single cells, the entire blob of cells was segmented as one (Fig 2c and Fig.2h - 2i).” This is because of the waxy hydrophobic cell surface, which makes mycobacterial cells form clumps/aggregates. The setup does, however, not have any problem segmenting and tracking individual mycobacteria when they are sparse (Fig. 2d and 2e). The abstract has been rephrased to avoid a misleading statement.

- It is not clear from the current dataset provided how the choice is made for defining the concentration of antibiotic for measuring resistance. For example, 1X MIC was used in case of INH against BCG whereas 10X MIC was used for Rifampicin. Will this have to be determined empirically or is this dependent upon the mode of action of antibiotics?

Reply: (Essentially the same as the question 5 asked by #Reviewer 1.)

In the original manuscript, we used antibiotic concentrations related to the MIC of wildtype and typically not the higher critical concentrations, which would give faster responses but may also impact resistant isolates, which we did not have access to then. For example, critical concentrations (CCs) of RIF, EMB, and LZD in liquid culture by WHO for *M. tuberculosis* are 1 mg/L, 5 mg/L and 1 mg/L, respectively¹⁻³, whereas the MICs are 0.06 mg/L, 2 mg/L, and 0.5 mg/L, respectively (this study). Thus using CCs could have led to a misleadingly fast impact.

In the revised version, we have had access to susceptible and resistant *M. bovis* BCG Russia strains. We added the results from pDST measurements on those new strains at the CCs for *M. tuberculosis* (Page 12, Line 8 - 18; Page 13, Line 1 - 16).

With respect to the specific question about RIF, we started with 1X MIC at 0.06 mg/L and observed the growth difference after about 8 hours (Supplementary Figure 2b). We then increased the RIF concentration to 10X MIC (0.6 mg/L, still lower than the CC for RIF at 1 mg/L) and observed the growth difference in under 3 hours (Figure 3a and Supplementary Figure 2a).

Also, how does the technique work when analyzing a heteroresistant isolate.

We have added experiments to illustrate the potential for detecting heteroresistant infections (see Answer 6, Reviewer 1) (Page 13, Line 18 - 28; Page 14, Line 1 - 12). The results showed that it is possible to detect heteroresistant infections because the method can track the growth of single microchambers and potentially single cells in microchambers.

- The workflow in the current form addresses the drug susceptibility testing and speeds it up but is still reliant on the culture of the inoculum to feed into the microfluidic chip. Can this approach be directly used on sputum samples itself, combining it with bacterial species identification as well, for example using the mycobacteria-specific trehalose dyes.

Reply: This is a great idea and likely a future perspective for our work. As mentioned in the manuscript (Page 4, Line 6 -7), the workflow of our fast pDST started from liquid culture. However, the method could start from sputum samples. Once the sputum is pretreated with mucolytic and decontaminant solutions (e.g. NaOH, Trisodium citrate, and N-acetyl L-cysteine) and resuspended in a growth medium, mycobacterial cells could be loaded onto the microfluidic chip for pDST and followed by species identification (ID)/detection using trehalose dyes (e.g. 3HC-3-Tre dye used in this work). The proposed workflow might be inoculum-independent but dependent on numerous other factors such as the growth stage of mycobacteria in sputum samples, the efficacy of the sputum pretreatment process, etc. Overall, the current work provides a strong foundation for combining ID and pDST directly from sputum samples using a microfluidic chip, microscopy, and advanced image analysis. We have added a sentence in the Discussion for this future perspective (Page 15, Line 31 - 32).

- What is the nature of the mutation in RIF E3? The MIC of the strain is reported to be 50 mg/L, yet it seems to be inhibited by exposure to RIF 10 mg/L.

Reply: We have added mutation information for the resistant *M. smegmatis* strains in the materials and methods (Page 16, Line 13 - 16). RIF E3 harbored a point mutation (S490L (TCG→TTG)) in the gene ortholog to the *rpoB* gene of the *M. tuberculosis* H37Rv strain. The level of resistance of RIF E3 is lesser than RIF E1 and RIF E2 because they harbored additional mutations in other genes (*i.e.* *cypX* or *sugC*) besides mutations in *rpoB*⁸.

References

1. (gtb), G. T. P. Technical Report on critical concentrations for drug susceptibility testing of medicines used in the treatment of drug-resistant tuberculosis. <https://www.who.int/publications/i/item/WHO-CDS-TB-2018.5> (2018).
2. (gtb), G. T. P. Technical Report on critical concentrations for drug susceptibility testing of isoniazid and the rifamycins (rifampicin, rifabutin and rifapentine). <https://www.who.int/publications/i/item/9789240017283> (2021).
3. (gtb), G. T. P. Technical manual for drug susceptibility testing of medicines used in the treatment of tuberculosis. <https://www.who.int/publications/i/item/9789241514842> (2018).
4. Schön, T. *et al.* Antimicrobial susceptibility testing of Mycobacterium tuberculosis complex isolates - the EUCAST broth microdilution reference method for MIC determination. *Clin. Microbiol. Infect.* **26**, 1488–1492 (2020).
5. Kandavalli, V., Karempudi, P., Larsson, J. & Elf, J. Rapid antibiotic susceptibility testing and species identification for mixed samples. *Nat. Commun.* **13**, 1–8 (2022).
6. Mistretta, M. *et al.* Dynamic microfluidic single-cell screening identifies pheno-tuning compounds to potentiate tuberculosis therapy. *Nat. Commun.* **15**, 4175 (2024).
7. Danchuk, S. N., McIntosh, F., Jamieson, F. B., May, K. & Behr, M. A. Bacillus Calmette-Guérin strains with defined resistance mutations: a new tool for tuberculosis laboratory quality control. *Clin. Microbiol. Infect.* **26**, 384.e5–384.e8 (2020).
8. Maeda, T., Kawada, M., Sakata, N., Kotani, H. & Furusawa, C. Laboratory evolution of Mycobacterium on agar plates for analysis of resistance acquisition and drug sensitivity profiles. *Sci. Rep.* **11**, 15136 (2021).

RESPONSE TO REVIEWER COMMENTS

Reviewer #1 (Remarks to the Author):

No further comments

Reply: we thank the reviewer for their time.

Reviewer #2 (Remarks to the Author):

The revised version of the manuscript by Elf and colleagues addresses my previous concerns, and I consider the manuscript suitable for publication in its current form.

Reply: we thank the reviewer for their kind words and recognition of our work's innovation.

Reviewer #3 (Remarks to the Author):

The authors have revised the manuscript incorporating the feedback from the previous reviews and have also included additional data on two other antibiotics, drug resistant strains and monitoring of heteroresistance. These revisions have strengthened the manuscript significantly and I have no other major comments.

- Supplementary information File, Page 3 Ln 17 "We used ports 2.1 and 2.2 to supply media without and without antibiotics." should be "We used ports 2.1 and 2.2 to supply media with and without antibiotics."

Reply: We thank the reviewer for the positive comments and suggestions to strengthen the method. We have modified the text in the Supplementary information as suggested. "We used ports 2.1 and 2.2 to supply media with and without antibiotics." (Page 3 Ln 18 -19).